# iENDEAVORS: Development and Testing of Virtual Reality Simulations for Nutrition and Dietetics

**DOI:** 10.3390/ijerph22091389

**Published:** 2025-09-05

**Authors:** Virginia Quick, Barbara Chamberlin, Devon Golem, Pinkin Panchal, Sylvia Gabriela Phillips, Carol Byrd-Bredbenner

**Affiliations:** 1 School of Environmental & Biological Sciences, Rutgers University, New Brunswick, NJ 08901, USA; pinkin.panchal@rutgers.edu (P.P.); bredbenner@sebs.rutgers.edu (C.B.-B.); 2Innovative Media Research and Extension, New Mexico State University, Las Cruces, NM 88003, USA; bchamber@nmsu.edu; 3Performance Nutrition and Dietetics Graduate Program, Arnold School of Public Health, University of South Carolina, Columbia, SC 29208, USA; dgolem@mailbox.sc.edu; 4Dietetic Internship Graduate Program, College of Agricultural, Consumer and Environmental Sciences, New Mexico State University, Las Cruces, NM 88003, USA; gabyphil@nmsu.edu

**Keywords:** virtual reality, dietetics, health education, nutrition, simulations, nutrition counseling, dietetic students, medical nutrition therapy, clinical nutrition, health professionals

## Abstract

Virtual Reality (VR) simulations provide immersive, realistic educational experiences that are increasingly used to enhance teaching and learning in nursing and medicine; however, use in dietetics lags. To fill this gap, four Nutrition Counselor VR simulations were developed collaboratively with the goal of building confidence in dietetic students’ nutrition counseling skills. After formative testing, pilot testing, and refinements, simulations were field tested with 34 dietetic students (91% women; age 25.67 ± 3.79 SD years; 68% White) from four supervised practice programs using a standard protocol administered by trained researchers (N = 5). Students completed a pre-survey, one VR simulation (≥2 times w/varying outcomes), and a post-survey. Online pre- and post-surveys examined changes in nutrition counseling skills, knowledge and self-efficacy, and comfort in using nutrition counseling skills. Paired *t*-tests revealed significant (*p* < 0.05) mean differences in nutrition counseling skill self-efficacy (medium effect size, *d* = 0.46) and comfort in using nutrition counseling skills (large effect size, *d* = 0.96) between the pre- and post-survey. At post-survey, >75% agreed the simulations helped build their nutrition assessment skills (79%) and counseling skills (88%) and prepared them to work with real patients (97%). Findings suggest the Nutrition Counselor VR simulations provided a realistic and safe learning environment that may be a valuable learning tool for dietetic students.

## 1. Introduction

Virtual Reality (VR) uses technology (e.g., headsets, hand controllers) to provide immersive, realistic simulated experiences that “transport” users to new situations and environments. In training future health professionals, VR provides opportunities to experience a virtual world comprising patients with whom learners interact, like in the real world, and expand their knowledge, decision-making abilities, and clinical and teamwork skills [1,2,3,4,5,6]. VR simulations offer a safe and supportive learning environment that delivers standardized, repeatable, on-demand clinical training with immediate, personalized feedback that builds skills and expertise while keeping patients safe from unintentional harm by novices [2,5,7]. Additionally, VR simulations facilitate self-paced learning, which is particularly beneficial for students requiring additional reinforcement or those enrolled in remote education programs [8].

A growing body of literature supports the effectiveness of VR in improving student learning outcomes in healthcare education [6,9,10,11,12,13,14,15,16,17,18,19,20,21,22]. Available research indicates that VR clinical simulations have improved health professionals’ application of theoretical knowledge in clinical settings, patient communication skills, teamwork, feelings of empathy for individuals from different backgrounds (e.g., cultural, socioeconomic, religious) than their own, understanding of patients’ life experiences and frames of reference, critical thinking and clinical decision-making skills, and knowledge and confidence in helping patients, as well as reduced anxiety when performing clinical procedures and lowered rates of incorrect decisions and errors [2,6,9,10,11,12,13,14,15,16,17,18,19,20,21,22,23,24,25,26,27]. Healthcare students trained using VR also were able to perform procedures nearly twice as proficiently as those who received traditional training [28].

VR simulations are increasingly used to enhance teaching and learning in future nursing and medicine professionals [3,4,5,10]. However, the current standard delivery of nutrition and dietetics education (i.e., didactic instruction) lags behind in the use of VR compared to other health education fields despite its potential [29]. Unlike traditional education methods, such as lectures, case studies, and role playing, VR has the potential to allow nutrition and dietetics students to visualize complex concepts in three-dimensional space, interact with virtual patients, and practice nutrition counseling in realistic settings. VR programs offer safe ways to fail, giving students the ability to experience emotional reactions to virtual patients, and ways to practice different responses, which may lower their anxiety and improve their practice with real patients. Thus, VR simulations may be able to foster deeper comprehension and application of nutrition counseling skills, as well as expanding decision-making abilities and clinical and teamwork skills, as evidenced by other health education fields using VR [1,2,3,4,5,6]. VR can simulate taking patient nutrition and health histories, conducting nutrition-focused physical exams, providing nutrition counseling and education using motivational interviewing, and placing a feeding tube. This, in turn, can effectively help students prepare for clinical practice and lessen strain on clinical practice supervisors by having better prepared students who master clinical skills more rapidly. By presenting students with a diverse range of simulated patient profiles—including individuals with varying backgrounds, levels of trust in healthcare providers, medical conditions, and treatment needs—VR may provide a more comprehensive learning experience. Furthermore, VR simulations offer real-time feedback, helping students correct errors and improve their communication strategies before encountering real patients [6]. Beyond academic advantages, VR simulations address critical challenges in clinical training, such as limited access to patient interactions and variability in case exposure.

The widespread adoption of VR simulations in dietetics education is hindered by several challenges. High implementation costs, including software development and hardware acquisition, remain a significant barrier [30]. Additionally, faculty require training to effectively integrate VR simulations into curricula, and technical issues may pose obstacles for students and faculty unfamiliar with the technology [30]. Compounding these obstacles is the limited body of research assessing the effectiveness of VR-based instruction on the development of professional competencies and its downstream impact on patient care outcomes in nutrition and dietetics education. To our knowledge, no other VR simulations have been developed and tested for their effectiveness in the dietetics education of students. These gaps highlight the need for innovative, evidence-informed approaches to health education that align with broader efforts to promote well-being and advance public health.

To begin to address these challenges, the Enhancing Nutrition and Dietetics Education with Artificial Intelligence and VR Simulations (iENDEAVORS) team developed VR simulations designed to build nutrition counseling skills. This 11-member team included experts in nutrition, dietetics supervised practice, dietetics education, instructional design, and virtual reality development. To evaluate the usability and educational value of these simulations, the development process included formative testing, pilot testing, and field testing. Thus, the purpose of this paper is to describe the design process of the VR simulations, revisions based on user feedback during formative testing, and field test findings on the feasibility and acceptability of VR-enhanced learning in dietetics education.

## 2. Materials and Methods

To ensure the VR simulations would be well accepted, evidence-based, and useful for dietetic students and educators, the design process was team-focused. This method allowed team members to apply and integrate their expertise to ensure the overall success of the project. Additionally, inclusion of all team members throughout the process helped create VR simulations that reflect the innovation of design professionals and current technology, pedagogical strength of educators, evidence-based practices of dietitians, congruence with requirements for dietetics supervised practice (i.e., Accreditation Council for Education in Nutrition and Dietetics [ACEND]; https://www.eatrightpro.org/acend (accessed on 28 August 2025)), and unique learning and teaching desires of dietetic interns and educators. (Note: ACEND is the accrediting agency for supervised practice programs leading to the Registered Dietitian Nutritionist [RDN] credential in the United States.)

### 2.1. Development and Formative Testing of VR Simulations

The VR simulation development and formative testing process took place over a 3-year period and had 4 main components. One component was identifying skills that are challenging to demonstrate and evaluate in dietetics supervised learning experiences (e.g., working with patients experiencing strong emotions, conducting nutrition focused exams). After lengthy discussion and consultation with preceptors who train dietetic students, the team generated a list of possible skills to develop, ultimately prioritizing 6: patient barriers/non-adherence, nutrition communication, nutrition counseling, assisting patients with emotional regulation, nasogastric tube placement, and nutrition focused physical exams. These 6 skills are essential for students to achieve ACEND competencies and are challenging to acquire through traditional methods. ACEND competencies associated with these skills served as the foundation for the VR learning objectives. Table 1 lists the skills addressed in the VR modules, learning objectives, and associated 2022 ACEND Competencies and Performance Indicators [31].

Once the skills were identified and learning objectives developed, the second component was to identify potential story lines for patient–dietitian VR simulation scenarios and emotional contexts that would effectively build dietetic students’ skills in communication (e.g., strategies such as eliciting genuine responses, handling confrontation, using clear language, answering medical questions, responding to patient emotional states such as fear, anxiety, anger, embarrassment), counseling (e.g., using motivational interviewing techniques, assessing readiness to learn and behavior change), and professionalism. The story line development also considered patient characteristics. Care was taken to ensure the simulated patients varied in age, race/ethnicity, sex, and pathology and would have realistic diet-related issues and concerns. Scenarios were set in both inpatient and outpatient settings and designed to last 10 to 15 min. One of the dietetic educators on the team took the lead in drafting simulation scripts. Script development included creating foundational case studies using backwards design, researching current evidence-based nutrition intervention and counseling guidelines specific to each case study, outlining counseling responses, generating decision-tree maps, and recording scripts to reflect intended tone and emotion. Other team members iteratively refined the drafts of the simulation scripts to ensure they addressed the skills and learning objectives, were reflective of realistic patient–dietitian interactions, and could be effectively rendered as VR simulations.

The third and fourth components occurred simultaneously with the first and second components. The third component involved selecting the VR hardware for simulation development and delivery. Cost, availability, ease of building the software–hardware interface, simplicity of use by instructors and students, distribution possibilities, and ease of development guided the selection process. The commercially available Meta Quest platform (initially Oculus) was identified as the distribution media most suitable for the purpose of this project, which would allow users with Meta Quest headsets to download the VR programs from the Meta store. This provided easy distribution to the most commonly used equipment.

The fourth component involved development of the animated characters and virtual world for the simulations, as well as the VR interface for user interaction. An important goal of the project simulations was to engage users in the emotional situations inherent in dietetics counseling interactions, thus the team had initially sought to make the animated characters and environments as realistic as possible. However, the state of animations possible at the time of development did not result in photorealistic characters. Though the team could create and render 3D models with more realistic skin, hair, eyes and clothing, the overall feeling of the characters as determined by the research team still fell into the “uncanny valley”. This term represents the phenomenon of interacting with a digital or computer-generated character that may look and present a human but feels sufficiently “off” to cause user uneasiness and discomfort. While this type of 3D rendering is commonly used in medical VR programs, the goal of those programs trend towards realistic representations of the human body, not meaningful emotional engagement with individuals. Instead of striving for a high degree of photorealism, the team opted to explore different types of characters and art rendering styles to facilitate a more emotional connection between the user and the VR patients. The team recorded a simple, short audio recording of a woman who is discouraged by her diabetes care and was crying through the frustration. Then, to accompany the audio recording, the team created original, animated human models with exaggerated eyes, eyebrows, and hands to enable greater manipulation of these features to convey emotion. After formative testing several sample character animations with the research team to determine potential user acceptance, the team finalized an artistic design and cell-shaded animation style that was thought to deliver a strong emotional impact, while still feeling personal and realistic. Once the character art style was established, the team developed environments (e.g., doctor’s office, hospital room) to match the style of the character and learning objectives of the simulation. Both inpatient and outpatient healthcare settings were used. All artwork was originated and completed by the development team, who created the models, textures, and animations.

Notably, each patient animation was hand-animated, meaning the lips and eyebrow movements, hand gestures, and eye activity were designed and created by animators. Though hand animation is time consuming, this approach was selected against a more economical approach of moving the lips and eyes in computer or AI generated manner when a character’s audio file is played. In hand-animating the interactions, the emotional impact and perceived realism of the characters was communicated to students, engaging them in more powerful ways. Animating each character for each scripted audio file provided greater manipulation of each character’s personal attributes for the desired emotional impact. Similarly, the team produced each audio file, directing voice artists in inflection, timing, and pronunciation to achieve the most realistic effect. The program website provides images of the four characters for viewing as well as a short video clip as an example of the user experience (https://nutritioncounselorvr.com/ (accessed on 28 August 2025)).

Each of the four scripts were an interaction between a registered dietitian nutritionist (the user’s frame of view) and a patient. Each simulation is branched so there are several versions of the interaction and different possible outcomes. The prompts give users the opportunity to make different decisions at specified points, which lead to different patient behaviors and endings each time the simulation is used. The extensive branching of the scripts allows users to repeat the simulations and learn from positive and negative interactions with the patients. These multiple reactions are scripted, and are not generated in real time by an AI system. Some options are ideal and demonstrate client-centered approaches that elicit a positive response from the patient. Other options directly conflict with counseling guidelines and lead to consequences revealed by the patient’s negative responses. Additionally, some options provide the opportunity to recover favorably from a previously selected negative option. Summative insights are provided along each pathway in a narrator’s voice to help the user learn from the interactions and responses they witness.

Extensive branching within the scripts also allows for a longer counseling session to be segmented into manageable learning sessions for users. For example, one of the scripts addresses barriers to numerous dietary behavior changes. The user may opt to focus on one of these behavior changes and select from the subsequent options related to that dietary behavior. Each simulation script has one to three outcomes with numerous pathways depending on the selected options. These pathways provide new learning experiences designed to increase engagement and critical thinking skills. Thus, users could repeat the VR simulations multiple times and by selecting different responses to prompts, have a novel learning experience.

The team worked through different scenarios for allowing the user to select an option, hear the RDN vocalize that option, and then hear the response of the patient. Simulation options were selected using hand controllers or the user’s own hands (i.e., enabled hand tracking feature). At the outset of VR development, hand controllers were used to interact in the VR simulation. Later, in the development phase, a hand tracking feature became available and was implemented to make it easier for users to choose prompts in the simulations. All narrated text is displayed on the screen to enhance accessibility for those with hearing disabilities.

For each simulation, educational profiles were developed for users and instructors. Each profile includes information on the target audience, purpose and description of the simulation, learning objectives, clinical and counseling topics to review before the simulation, references to support the pre-simulation review, notes for the instructor regarding possible outcomes and pathways, an electronic medical record for the patient, and debriefing discussion topics and questions to help users reflect on the experience and gain further insights. A description of the VR simulations is found in Table 2. More information on the Nutrition Counselor VR app can be found on the study website: https://nutritioncounselorvr.com/.

### 2.2. Pilot Testing of VR Simulations

Pilot testing was performed to further improve the VR simulations prior to field testing. The pilot testing protocol was approved by the Institutional Review Board at the authors’ institutions. All participants gave informed consent electronically prior to study implementation.

Beta versions of the simulations were pilot tested with 18 dietetic interns (100% enrolled in a supervised practice program) from two institutions in different geographic regions (northeastern, southwestern) in the United States to identify needed refinements to the scripts, characters, settings, and overall simulation experience. A study recruitment announcement was sent electronically to dietetic interns at these supervised practice programs explaining the purpose of the study, time commitment, location of the testing, and stipend (USD 25 electronic gift card). After completing an electronic informed consent form, participants scheduled a time to visit the campus to complete a randomly assigned VR simulation with the researcher. On campus, each intern completed one VR simulation (up to 3 times, encouraged to choose different prompts on their own each time) during a 20 to 30 min session. Prior to the VR experience, the researcher explained the purpose of the experience and gave a brief orientation on the VR equipment (i.e., Oculus^®^ Quest VR headset, versions 2 and 3r). During the VR experience, interns were seated and listened to and interacted with the patient. The researcher remained nearby to provide guidance as needed. Interns proceeded through the simulation at their own pace and chose prompts as they desired. Within 24 h after the simulation, each intern completed a 30 min virtual interview conducted by a trained moderator using a semi-structured interview guide [32]. All interviews were recorded, transcribed, and content analyzed by two trained researchers to identify common themes.

The moderator began the interview by asking the intern to take a few minutes to think about the overall simulation experience—including the orientation. Most interns (~90%) reported having little or no prior experience with VR and some stated they were nervous or fearful of the technology at first, but afterwards felt it was a “cool” and valuable learning experience. Most thought having an opportunity to become comfortable with the technology—as stated by one student, “tutorials on how to use the equipment for VR”—would have been helpful. A few suggested having a demonstration of the VR equipment would be helpful, too. Most felt having a brief overview of the patient they would be counseling in the VR simulation would better prepare them for the experience. Students suggested providing information on the patient’s background similar to what a clinician would view in an electronic medical record. When asked how prepared they felt they were to counsel someone with the patient’s disease state, most students felt comfortable going into the simulation due to their prior education and training. However, a few felt nervous about the counseling aspect because they had never counseled a real patient before and/or did not know what to expect.

The next part of the interview explored students’ perceptions about the simulation itself (i.e., patient and healthcare setting). When asked about the patient in the simulations, students felt patients were believable and like those they might encounter in real-life settings. As one student stated, “The patients were believable, like the ones I have seen in my training, especially when you are talking about things that are personal, like weight, or scary, like getting an NG tube placement.” Another commented, “The voices of the characters seemed realistic…I could hear from Samuel [the simulation patient] his tone and feel his emotion even though he was not a real person.” When asked about their thoughts on the setting of the simulations, a few felt like the in-patient setting did not seem like a typical hospital they had worked in while others felt the setting seemed like a real hospital. Some suggestions to make the setting feel more real were to add more machines/equipment (e.g., intravenous drip in the room), curtains for a shared patient room, and a full food tray to show that the patient was not eating. For the outpatient setting, most felt the setting was exactly what they would experience. “The round table, like sitting at the little round table, was exactly what it looked like in my rotation [supervised practice],” one student stated.

Next, students’ perceptions of the flow and length of the simulation were explored. Students were positive about how the simulation began and progressed to the end; as one student commented, “The flow seemed exactly what you would expect…” As for the length, most felt the time needed to complete the simulation was just about right. As one student stated, “It wasn’t too long where you would be antsy since you are sitting, but long enough to get what you needed out of it. I felt engaged throughout, especially when I was able to select the response to prompts.”

The interview also explored perceptions of simulation quality. Some students wanted better instructions on how to physically manipulate the assets in the simulations (e.g., the user’s virtual tablet that was mainly used to select responses to prompts) either through narration or easy-to-read text embedded in the simulation. Some also wanted more interaction opportunities during the simulation (e.g., more prompts to select). Several students reported technology “glitches” (e.g., pauses or skips) and/or had difficulty completing physical tasks in the simulation, such as holding objects (e.g., tablet) or selecting the prompts. Some students felt that, during the nutrition focused physical exam scenario, the dietitian used medical terminology (e.g., orbital region, trapezoid) that would not be understandable to most patients. In terms of sound, some commented that they liked the background noises (e.g., “beeping” sounds) because these are typical in a hospital setting. The only sound issues were related to some of the glitches in the initial programming of the beta version of the software (e.g., character mouths moving but no talking).

In terms of the value and usefulness of the simulations, students reported enjoying the challenge of counseling the difficult patients and being able to practice in a safe learning environment that reinforced their motivational interviewing skills. One student commented that “VR can be used to give a safe environment for students to fail. It saves a lot of resources and is very flexible timewise to use VR, so there is a lot of value to it.” Another stated, “with VR, you don’t know what the situation is going to be, including the dialogue beforehand, so it challenges you to think on your feet. Even though VR is animated, it is quite helpful for students to use and learn from their mistakes.” Students rated their motivation to use the knowledge and skills gained from the simulation as 4 on a 5-point scale (higher scores reflect greater motivation).

When asked about how the simulation affected their nutrition counseling skills, about three quarters felt more confident in their nutrition counseling skills after the VR experience. As one student stated, “It can be overwhelming when you initially meet a patient for the first time, so I am feeling more confident knowing I had this experience. I will be prepared mentally that I can be in such situation in knowing how to handle it.”

To better understand how instructors/educators could build on the simulation experience to make it even more valuable, students were asked to recommend post-simulation activities. A common suggestion was to add facilitated debriefing sessions post-simulation. One student stated, “It would be good to have a debriefing with other dietetic students to learn from each other.”

The interview concluded by asking students for additional suggestions for improving the VR simulations as well as ideas to make them more appealing to dietetic students. Most felt using VR was already appealing; however, a few felt it would be nice to have the user interact more via more prompts and options. Some suggested having more patient simulation scenarios with different health conditions.

Thus, results from the pilot testing led the team to resolve identified technological glitches. Issues related to completing physical tasks were resolved by developing more complete instructions for simulation use as well as switching to the hand tracking feature. Additionally, supporting materials for educators and dietetic students that provide preparatory information on each patient, known as the educational profile, were developed. The educational profile states the purpose, description, and learning goals of the simulation and provides the electronic medical record of the patient. The profile also includes educator resources for using the simulation for instruction, along with a debriefing guide with suggested questions to facilitate discussions post-simulation. Additionally, pilot testing findings led to enhancements in the flow and language used in simulation scenarios to further improve the overall VR experience.

### 2.3. Field Testing VR Simulations

The field test also was approved by the Institutional Review Board at the authors’ institution. Like the pilot test, all participants gave informed consent electronically prior to participating.

#### 2.3.1. Recruitment

Dietetic students from four supervised practice programs located in the northeastern United States were recruited to participate in the VR simulation field test. The study recruitment announcement was sent electronically to dietetic students at these supervised practice programs explaining the purpose of the study, time commitment, location of the testing, and stipend (i.e., USD 100 e-gift card) for participating.

#### 2.3.2. Field Test Design

The field test design included review of the final release version of the software. While modern software is usually updated frequently once released, the version used in this field test study was considered the final candidate, ready for release in stores. The study design involved a pre-survey, participation in the intervention, and a post-survey. Dietetic students expressing interest in participation completed the online informed consent and pre-test survey 7 to 10 days before using the simulation. Students were systematically randomized to one of the four simulations in the order in which the consent and pre-test survey were completed. All field testing was conducted at Rutgers University, New Brunswick campus. The online post-survey was administered immediately after completing in the simulation debriefing session.

The protocol for field testing the simulations was developed and implemented by research assistants (N = 5) trained to execute it with fidelity. Groups of 2 to 4 students systematically randomly assigned to the same simulation simultaneously, but independently, participated in the simulation. Groups were all in the same spacious room and were physically separated to prevent distractions. At the beginning of the field test, research assistants spent about 10 min explaining the field test purpose, providing preparatory materials and instructing students to individually review them (i.e., educational profile and the Nutrition Counselor VR website (QR code used to access site on their own electronic device)). The educational profile included an electronic medical record that a clinician would review prior to counseling patients. Next, students were oriented to the VR headset and taught how to proceed through the simulation using the hand tracking feature. Research assistants informed students that during the simulation they would be sitting, listening (not speaking), and interacting using their hands in the VR space at their own pace and that research assistants would be close by to provide guidance as needed. Students also were instructed to complete the simulation again after finishing (up to 3 times or up to 30 min total) and to select different responses to the prompts each time. In an effort to avoid cybersickness, participants had to be seated during the VR simulation and were limited to a maximum of 30 min. Participants were instructed to stop at any time if they felt dizzy or nauseous.

After completing the simulation, a trained facilitator conducted a 15 to 20 min debriefing session with 7 to 8 questions tailored to the simulation. Students then completed the online post-test survey immediately after the debriefing session.

#### 2.3.3. Instruments

All items on the pre- and post-test surveys were created de novo, except for items related to nutrition counseling skill self-efficacy, which were modified from the valid, reliable Dietitians’ Counseling Self-Efficacy Scale [33]. Each survey took about 20 min to complete and assessed nutrition knowledge, nutrition counseling skill self-efficacy, perceived comfort using nutrition counseling skills, and attitudes toward RDN nutrition counseling responsibilities. The pre-survey also collected sociodemographic information. The post-survey also evaluated perceptions related to the use of the simulations.

Two multiple choice items focused on knowledge-related questions specific to each simulation (8 total) were developed for the Knowledge scale. Items were scored as correct (1 point) or incorrect (0 points). Scale scores were determined by summing item scores.

For the Nutrition Counseling Skill Self-Efficacy scale, a total of 15 items from the original 25-item Dietitians’ Counseling Self-Efficacy Scale [33] were adapted for this study by using a 5-point Likert scale (i.e., 1 = not confident at all, 2 = slightly confident, 3 = somewhat confident, 4 = quite confident, 5 = very confident) to assess confidence in using various nutrition counseling skills (e.g., use nonverbal cues to convey empathy, warmth, and support; elicit more elaborate responses; define with patient clear and concise goals that are measurable and behavior-focused). Item scores were averaged, with higher mean scores indicating greater confidence in nutrition counseling skills.

For the Comfort Using Nutrition Counseling Skills scale, a total of 8 items (2 per simulation) evaluated how comfortable students felt performing certain nutrition counseling skills specific to the simulations (e.g., counseling patients feeling overwhelmed by their medical treatment, counseling angry or upset patients, performing nutrition-focused physical exams, providing patient education on blood glucose monitoring). Responses were on a 5-point Likert scale (1 = very uncomfortable, 2 = uncomfortable, 3 = neither comfortable nor uncomfortable, 4 = comfortable, 5 = very comfortable). Item scores were averaged, with higher mean scores indicating greater comfort in using nutrition counseling skills.

For the Attitudes Toward RDN Nutrition Counseling Responsibilities scale, eight items (2 per simulation) assessed student attitudes toward RDN responsibilities for providing nutrition counseling related to the simulations (e.g., “RDNs must be competent in counseling patients with diabetes about how to self-monitor their blood glucose”; “It is important for me, as a future RDN, to learn the process for placing a nasogastric tube”; “RDNs must be competent in performing a Nutrition-Focused Physical Exam”). Item scores were averaged, with higher mean scores indicating more positive attitudes toward RDN nutrition counseling responsibilities.

Students reported sociodemographic characteristics such as their age, biological sex, gender orientation, race/ethnicity, education level, prior clinical dietetic work experience, and onsite (professional workplace) supervised practice hours completed so far on the pre-survey. Students also reported prior experience using VR and any visual or hearing impairment (e.g., color blind, corrective lens) that would need to be considered in the simulation experience.

During the post-test survey only, students were asked to report how they felt before (7 items; e.g., “I felt prepared to counsel real patients”; “I understood the purpose and objectives of the simulation”), during (5 items; e.g., “I was given enough information to complete the simulation”; “I felt the simulation facilitated my ability to independently problem solve”) and after (8 items; e.g., “I felt the simulation required more medical nutrition therapy knowledge than I have”; “The simulation helped me build my nutrition counseling skills”) the simulation experience. All items were answered using a 5-point Likert agreement scale (i.e., 1 = strongly disagree to 5 = strongly agree). Seven other items, using the Likert agreement scale, evaluated the overall simulation experience (e.g., “The patient scenario felt realistic”; “The simulation was a helpful learning experience”). Item scores on each scale were averaged, with higher scores indicating a more positive experience. An additional item asked students to indicate who they believed would benefit from using the VR simulations.

### 2.4. Data Analysis and Power Analysis

Descriptive statistics were performed for all items in the pre- and post-test surveys. Internal consistency scores for baseline measures with more than 2 items, except the Knowledge scale where the Kuder-Richardson Formula (KR-20) was calculated, were conducted using Cronbach’s alpha. Paired *t*-tests compared pre- and post-survey changes in knowledge, nutrition counseling skill self-efficacy, comfort in nutrition counseling, and attitudes towards RDNs nutrition counseling responsibilities. The Benjamini and Hochberg procedure (also known as the False Discovery Rate) was used to correct for multiple comparisons in the paired *t*-tests performed. Cohen’s *d* examined effect sizes (0.20 = small; 0.50 medium; 0.80 large) [34]. All data analyses were conducted using IBM SPSS (version 29) with a significance level set at *p* < 0.05. An a priori power analysis was conducted using G*Power version 3.1.9.7 to determine the minimum sample size required to test the study hypothesis. Results revealed the required sample size to achieve 80% power for detecting a medium effect at a significance criterion of α = 0.05 was N = 34 for paired *t*-tests (two-tailed).

## 3. Results

A total of 34 dietetic students participated in the field test, which was adequate to test our study hypothesis. Table 3 reports their sociodemographic data. The sample was fairly diverse in race/ethnicity with almost 12% identifying as Hispanic or Latino and about two-thirds identifying as White (68%). The sample was mainly women (91%), which reflects the current demographics of the nutrition and dietetic profession. Over half of the students had a master’s degree (59%), which was due to the structure of the dietetic supervised practice program for 20 of the participants, which requires dietetic students to complete their master’s degree prior to starting their supervised practice. The type of dietetic supervised practice most were enrolled in was a dietetic internship (82%), with the remainder enrolled in a similar, but more competency-based program. About half had prior clinical dietetics work experience. Dietetics supervised practice hour accrual averaged about 485 h, with the average professional worksite practice hours being about 300 (range 0 to 1000) and about 187 h in simulated (but not virtual reality) experiences (e.g., case studies, manikins). Most students reported English was the primary language they spoke (91%) and read (94%). About half (47%) reported wearing corrective lenses in the form of eyeglasses, over one-third (38%) wore contact lenses, and 27% wore both eyeglasses and contact lenses. No one was colorblind or had a hearing impairment. Most participants had limited experiences using VR; half (53%) had never used VR before, and one-third (35%) reported using VR one or two times before.

The Kuder–Richardson 20 (KR-20) internal consistency for the knowledge scale was 0.89 indicating strong reliability (Table 4). Cronbach alpha coefficients for the Likert scales also were good, ranging from 0.77 to 0.89. Paired *t*-tests revealed significant increases in Nutrition Counseling Skill Self-efficacy scores (medium effect size) and Comfort in Using Nutrition Counseling Skills (large effect size) between the pre- and post-survey even after adjusting *p*-values to control for multiple comparisons. There were no significant changes in Knowledge or Attitudes toward RDNs Nutrition Counseling Responsibility scales between pre- and post-survey.

Student perceptions of the simulation experience at post-survey were favorable (Table 5). That is, at the beginning of the simulation the vast majority agreed they felt prepared for the simulation. For instance, they felt prepared to counsel patients, complete the simulation, were encouraged to complete the simulation, understood the goals of the simulation, and had sufficient information to fully engage with the simulation. Less than one-third were concerned (felt nervous) about using VR technology. During the simulation, almost all students agreed that the simulation provided the assistance needed to complete the simulation (e.g., they had enough information, cues promoted their ability to proceed, simulation facility independent problem solving, simulation encouraged exploring options in the simulation). Students also reported they felt the instructor (i.e., research assistant who were available during the implementation of the field testing) was responsive to their needs during the simulation. After the simulation, nearly all agreed the simulation felt realistic, using the simulation was helpful and enjoyable, and felt the simulations helped to build their nutrition assessment and nutrition counseling skills. When asked at what stage they thought dietetics students would benefit from using the VR simulations, most felt that dietetic students early in their undergraduate (pre-supervised practice) program and/or graduate dietetic students in their supervised practice programs prior to starting clinical supervised practice experiences would benefit the most.

## 4. Discussion

The purpose of this paper was to describe the development and formative testing of dietetics VR simulations designed to enhance dietetic students’ nutrition counseling skills, and the pilot and field testing of these simulations. This is one of the first studies to apply VR simulations in dietetics education, building on a limited base [8,29]. Findings from both pilot and field testing provide support for the usefulness, acceptability, and potential educational value of these VR simulations in dietetics education. Specifically, dietetic students who participated in the VR simulation field testing experienced significant increases in nutrition counseling skill self-efficacy and comfort in using counseling skills. These findings suggest that immersive, interactive VR experiences may be helpful in building students’ confidence and comfort in applying nutrition counseling skills in challenging patient interactions.

The observed increases in nutrition counseling self-efficacy and comfort using nutrition counseling skills are noteworthy given that self-efficacy is a known predictor of behavior change, per social cognitive theory [33,35]. That is, dietetic students who feel more confident in their ability to counsel patients are more likely to engage in and successfully carry out such tasks during their supervised practice training and early professional careers. Additionally, the instructor guidance during and after the simulation was viewed as being responsive and supportive to the dietetic students. Thus, VR simulations created a psychologically safe and supportive environment for students to practice their nutrition counseling skills, reinforcing the idea that realistic but low-stakes clinical practice can foster skill development and readiness for real-world application [6].

Despite improvements in nutrition counseling self-efficacy and comfort using nutrition counseling skills, knowledge scores did not significantly change at post-survey. This may be due, in part, to the high knowledge scores at baseline leaving little room for improvement. Additionally, knowledge was assessed by only two items per simulation, which could not assess the breadth of knowledge associated with the clinical condition in the simulation. It is also possible that the primary mechanism of the simulations’ effectiveness lies not in content delivery but in experiential learning—applying existing knowledge in realistic contexts to refine skills and decision-making, which was found to be the case in other intervention studies using VR simulations among nursing students [36]. Active learning methods such as that provided by VR simulations have previously been found to facilitate the development of logical reasoning [37]. The stability of attitudes toward RDN counseling responsibilities likely reflects a ceiling effect, as pre-survey scores were already high, suggesting that dietetic students entered the field test with an already high appreciation for the role of RDNs providing nutrition counseling skills in dietetics practice.

The dietetic students’ qualitative feedback immediately after using the simulation in the pilot study further illuminates the value of the simulations. Most found the patient scenarios realistic, emotionally engaging, and reflective of their supervised practice experiences. Participants appreciated the opportunity to make decisions, receive cues and feedback, and observe the outcomes of different choices. These aspects of simulation design likely contributed to the reported increases in engagement, critical thinking, and nutrition counseling skill development. Furthermore, the immersive nature of the VR simulations provided students with a safe space to practice handling difficult emotions, such as patient fear, frustration, or resistance—scenarios that can be especially intimidating for novice dietetic students. These findings are consistent with prior research demonstrating the effectiveness of VR in improving learning outcomes across health profession education, including communication, empathy, teamwork, and critical thinking [6,10,11,14,15,17,20]. Notably, healthcare students trained using VR perform procedures more proficiently and with greater confidence than those trained using traditional instructional methods [28]. However, VR use in dietetics education has remained relatively limited to date. This study helps bridge that gap by providing empirical support for the feasibility and utility of VR simulations in developing key dietetics education competencies among dietetic students.

The developed VR simulations also addressed known barriers to effective clinical training, such as variability in patient case exposure and limited opportunities for repeated practice [2]. The branched script design and multiple pathways allowed users to engage with different outcomes in subsequent uses of the same simulation, offering a more dynamic and individualized learning experience. Students also valued the post-simulation debriefing sessions by a trained facilitator, which provided additional opportunities to reflect, synthesize learning, and reinforce professional communication and evidence-based practices.

Relative to the development of additional VR simulations, if this project were started now, it is likely artificial intelligence (AI) would be considered to create models, support animation, or generate narration. However, it is worth noting that the majority of participants in this study regularly used terms such as “realistic” and “personal” in their feedback, and felt the simulations were “real”. The goal of these simulations was not to simply offer practice in skills or provide knowledge of environments, but to immerse users in the strong emotions of patients they will encounter. Automated scripting of interactions with AI generated patients could expand the scope of the simulations and create greater branching and variability of patient interactions. However, the value of this developed VR program may be in the emotional impact granted by interactions specifically created to move users to empathize with their clients. Additionally, the field of development is still identifying the costs of AI development in terms of accuracy and the required review of content, as well as legal and ethical implications. Consideration of the compromises of content diversity versus more emotionally impactful experiences is critical for developers of training simulations as they prioritize desired outcomes.

However, it is important to note study limitations. The small sample size limits generalizability of findings. Additionally, the lack of a control group in the VR field test limits the ability to determine if observed outcomes are truly due to the intervention. Furthermore, the short duration of the field test may not fully capture long-term impacts of the VR simulations on clinical performance. While the simulations reflected commonly faced and challenging scenarios in dietetics practice, they did not represent the full spectrum of clinical cases, which may constrain transferability. Finally, most students had little prior experience with VR technology, raising the possibility of novelty effects influencing their perceptions.

Despite these limitations, there are many important study strengths. The team-based approach in developing the VR simulations with experts in nutrition, dietetic supervised practice, dietetics education, instruction design and virtual reality interface developers helped to create realistic patient scenarios in various healthcare settings using the most appropriate VR platforms. The extensive, iterative formative testing that occurred during the development process ensured user acceptability of the simulations and should serve as a model for future simulation development. The careful consideration of pilot test results helped further improve the quality and acceptability of the simulations. Complementing formative and pilot testing with field testing is critical in assessing the potential impact. The high internal consistency of scales and rigorous simulation development and implementation processes performed is another important study strength.

Future research should explore the long-term impact of VR simulations on knowledge retention, clinical performance, and real-world patient care that impacts public health and promotion efforts. Larger-scale studies with diverse student populations across multiple programs will be critical to determine broader applicability. Moreover, examining how VR integrates with existing curricula and instructor facilitation strategies—such as debriefing and reflective journaling—will help identify best practices for implementation. Faculty development and institutional support also will be necessary to ensure sustainable adoption of VR simulations in dietetics education. Future work also may explore expanding the simulation case studies to include more complex clinical scenarios and interprofessional team-based interactions.

## 5. Conclusions

In conclusion, the VR simulations were well-received by dietetic students and were found to have meaningful improvements in nutrition counseling self-efficacy and comfort using nutrition counseling skills. The VR simulations provided a realistic, flexible, and psychologically safe learning environment that fostered engagement, reflection, and nutrition counseling skill development in dietetic students. Overall, VR can be a safe, repeatable, on-demand training tool for building counseling skills, especially when preparing dietetic students before supervised clinical practice. As the field of dietetics education continues to evolve, immersive VR experiences hold great promise for preparing future nutrition and health professionals with the skills, confidence, and adaptability needed to meet the demands of clinical nutrition practice. Broader adoption of this innovative educational strategy could enhance the quality and consistency of training while helping to address ongoing challenges in supervised practice education.

## Figures and Tables

**Table 1 ijerph-22-01389-t001:** Skills, objectives, and competencies addressed in VR simulations.

Skill	Learning Objectives	2022 FEM ACEND Competencies and Performance Indicators (C/PI) *	2022 ACEND Competencies for Dietetic Internships (CRDNs) *
Barriers and non-adherence	Describe strategies to help patients address barriers that lead to non-adherence. Provide examples of collaboration with patients to identify barriers to dietary behavior change. Explain strategies for addressing typical barriers associated with blood glucose monitoring and management. Describe how to support patients through all steps of addressing a barrier.	1.6 Applies knowledge of social, psychological, and environmental aspects of eating and food. 1.7 Integrates the principles of cultural competence within own practice and when directing services. 1.7.2 Applies knowledge of food eating patterns and food trends. 1.7.5 Applies culturally sensitive approaches and communication skills. 2.3 Utilizes the nutrition care process with individuals, groups, or populations in a variety of practice settings. 2.3.10 Determines barriers that might influence a client/patient’s nutritional status. 2.3.21 Assesses client/patient’s compliance with nutrition intervention. 2.3.22 Identifies barriers to meeting client/patient’s nutrition goals and makes recommendations to modify the nutrition plan of care or nutrition intervention and communicates changes to client/patient and others. 6.1.1 Considers multiple factors when problem solving.	2.7 Apply change management strategies to achieve desired outcomes. 3.4 Provide instruction to clients/patients for self-monitoring blood glucose considering diabetes medication and medical nutrition therapy plan.
Nutrition Communication	Explain the importance of listening intently to understand patient’s cognitions.	2.3.2 Interviews client/patient to collect subjective information considering the determinants of health. 2.3.11 Determines accuracy and currency of nutrition assessment data. 2.3.23 Summarizes impact of nutrition interventions on client/patient’s nutrition outcomes, considering client/patient-centered care. 2.3.24 Identifies, analyzes, and communicates reasons for deviation from expected nutrition outcomes. 2.4.10 Translates basic to advanced food and nutrition science knowledge into understandable language tailored to the audience. 7.2.1 Applies effective and ethical communication skills and techniques to achieve desired goals and outcomes.	3.7 Demonstrate effective communication and documentation skills for clinical and client services in a variety of formats and settings, which include telehealth and other information technologies and digital media.
Nutrition Counseling	Provide examples of collaboration with patients to establish individualized goals/objectives. Apply behavior change theories for nutritional health promotion and disease prevention. Discuss relevant aspects of nutrition assessment with patients. Summarize findings of patient’s nutrition assessment. Explain plan of care and next steps to patients.	2.4 Implements or coordinates nutritional interventions for individuals, groups, or populations. 2.4.7 Assesses the audience’s readiness to learn and identifies barriers to learning. 2.4.14 Applies counseling principles and evidence-informed practice when providing individual or group sessions. 2.4.17 Demonstrates awareness of various appropriate counseling techniques. 7.1.4 Applies client/patient-centered principles to all activities and services.	3.10 Use effective education and counseling skills to facilitate behavior change.
Emotional Regulation	Describe strategies for working with a fearful patient in flight mode. Describe strategies for working with an angry patient. Describe strategies for moving a patient from expressing overwhelm/defeat to addressing the health/treatment challenge.	7.1.1 Demonstrates ethical behaviors in accordance with the professional Code of Ethics. 7.1.6 Practices in a manner that respects diversity and avoids prejudicial treatment	2.10 Demonstrate professional attributes in all areas of practice.
Nasogastric Tube Placement	Discuss the purpose, need, benefits, and risks of nasogastric tube feedings with patients. Describe the steps involved in nasogastric tube placement with patients.	2.5.4 Considers client/patient factors, nutritional impact, indications, side effects, contraindications, benefits, risks, alternatives, and foundational sciences when prescribing, recommending, and administering nutrition related drug therapy.	3.5 Explain the steps involved and observe the placement of nasogastric or nasoenteric feeding tubes; if available, assist in the process of placing nasogastric or nasoenteric feeding tubes.
Nutrition-Focused Physical Exam	Describe the steps involved in a Nutrition-Focused Physical Exam.	2.3.3 Conducts a nutrition focused physical exam.	3.2 Conduct nutrition focused physical exams.

* A list of all 2022 ACEND Competencies and Performance Indicators by program type can be found elsewhere [31].

**Table 2 ijerph-22-01389-t002:** VR simulation description.

VR Simulation Description
Dolores: Set in an outpatient clinical setting, an older Hispanic woman with type 2 diabetes mellitus expresses overwhelm/defeat with her blood glucose management during her counseling session with the RDN. The RDN must navigate the conversation to help support and address this patient’s barriers to blood glucose management.
Craig: Set in an inpatient clinical setting, a middle-aged, White man with Crohn’s disease is angry and unhappy with his healthcare team interactions. The RDN must navigate a difficult conversation by using de-escalation techniques as well as perform a nutrition focused physical exam to assess for malnutrition.
Samuel: Set in an in-patient clinical setting, a middle-aged, Black man with swallowing difficulty expresses fear at having a nasogastric (NG) tube placement. The RDN must use de-escalation techniques when communicating to reduce this patient’s fear towards the NG tube placement and walks him through the procedure step-by-step.
Jhem: Set in an out-patient clinical setting, a single mom with a recent diagnosis of cardiovascular disease and hypertension has been struggling with weight loss management and making it to her scheduled appointments. The RDN must use motivational interviewing skills to help address Jhem’s barriers to dietary behavior change.

**Table 3 ijerph-22-01389-t003:** Sociodemographic characteristics of participants (N = 34).

Characteristic	N (%) or Mean ± SD
Age ^a^	25.67 ± 3.79
Sex (% female)	31 (91.2)
Gender (% female)	31 (91.2)
Race/Ethnicity	
Asian	5 (14.7)
Black or African American	1 (2.9)
Hispanic or Latino	4 (11.8)
Middle Eastern or North African	3 (8.8)
Native Hawaiian or Pacific Islander	0 (0.0)
White	23 (67.6)
Education Level	
Bachelor’s degree	14 (41.2)
Master’s degree	20 (58.8)
Program Type Enrolled	
Dietetic Internship ^b^	28 (82.4)
Future Education Model ^c^	6 (17.6)
Prior clinical dietetics work experience (% yes)	18 (52.9)
If yes, years total	1.58 ± 1.26
Hours of simulated clinical dietetics	187.07 ± 290.70
Hours of on-site clinical dietetics	298.24 ± 299.06

^a^ n = 33; one participant selected “over the age of 40 years” response and thus was not included in the average due to not having her exact age. ^b^ A dietetic internship is a supervised practical experience required for individuals seeking to become a registered dietitian nutritionist that requires a minimum of 1000 h in various settings like hospitals and foodservice operations. ^c^ A future education model program is a graduate-level program that combines coursework and supervised experiential learning to prepare individuals to become registered dietitian nutritionists and follows a competency-based approach.

**Table 4 ijerph-22-01389-t004:** Study Outcomes: Baseline and post measurement completers (N = 34).

Measure	# Items	Reliability Coefficient *	Pre-test Mean ± SD	Post-test Mean ± SD	Paired *t*-test ^c^		Cohen’s *d*
					*T* (*df*)	*p*	
Knowledge ^a^	8	0.89	6.47 + 1.05	6.26 + 1.26	1.04 (33)	0.607	0.18
Diabetes Management with Dolores Simulation (n = 9)	2	-	1.82 + 0.39	1.71 + 0.46	0.00 (8)	1.00	0.00
Malnutrition with Craig Simulation (n = 8)	2	-	1.56 + 0.61	1.24 + 0.61	0.00 (7)	1.00	0.00
Nasogastric Tube Placement with Samuel Simulation (n = 7)	2	-	1.65 + 0.49	1.76 + 0.50	1.55 (6)	0.459	0.59
Behavioral Weight Loss Management with Jhem Simulation (n = 10)	2	-	1.44 + 0.56	1.56 + 0.50	0.00 (9)	1.00	0.00
Nutrition Counseling Skill Self-Efficacy ^b^	15	0.89	3.92 ± 0.48	4.14 ± 0.46	2.69 (33)	0.044	0.46
Comfort in Using Nutrition Counseling Skills ^b^	8	0.84	3.44 ± 0.69	3.91 ± 0.58	5.59 (33)	<0.001	0.96
Attitudes Toward RDN Nutrition Counseling Responsibilities ^b^	8	0.77	4.61 ± 0.38	4.62 ± 0.49	0.10 (33)	1.00	0.02

* All reliability coefficients are Cronbach’s alpha, except Knowledge, which is a Kuder–Richardson Formula (KR-20). ^a^ Possible score 0 to 8 for total knowledge and 0 to 2 for individual simulation knowledge scores. ^b^ Possible score 1 to 5. ^c^ Paired tests with *p*-values adjusted using the Benjamini–Hochberg procedure to control for multiple comparisons.

**Table 5 ijerph-22-01389-t005:** Overall perceptions of the VR experience (N = 34).

Item	N (%) Strongly Agree or Agree	Mean ± SD
At the beginning of the simulation…		
Preparation for the simulation		4.53 ± 0.62
I felt prepared to counsel real patients.	29 (85.3)	4.09 ± 0.79
I felt prepared to complete the simulation.	32 (94.1)	4.47 ± 0.62
I had the directions to complete the simulation.	33 (97.1)	4.71 ± 0.52
I had the encouragement to complete the simulation.	32 (94.1)	4.59 ± 0.61
I understood the purpose and objectives of the simulation.	33 (97.1)	4.59 ± 0.66
I had enough information to fully engage with the simulation.	33 (97.1)	4.74 ± 0.51
Concern about using the simulation		
I felt nervous using VR technology.	10 (29.4)	2.56 ± 1.16
During the simulation…		
Assistance from the simulation		4.62 ± 0.58
I was given enough information to complete the simulation.	33 (97.1)	4.71 ± 0.63
The cues provided by the simulation promoted my ability to complete the simulation.	34 (100.0)	4.76 ± 0.43
The instructor recognized my need for help during the simulation.	33 (97.1)	4.29 ± 0.87
I felt the simulation facilitated my ability to independently problem solve.	32 (94.1)	4.47 ± 0.62
I was encouraged to explore all possibilities offered in the simulation.	34 (100.0)	4.85 ± 0.36
Assistance from the instructor		
The instructor recognized my need for help during the simulation.	33 (97.1)	
After the simulation…		
Preparation for the simulation		1.97 ± 0.97
I felt the simulation required more MNT * knowledge than I have.	4 (11.8)	1.94 ± 1.13
I felt the simulation required more advanced counseling skills than I have	4 (11.8)	2.15 ± 1.10
I felt the simulation required more nutrition assessment skills than I have.	0 (0.0)	1.82 ± 0.67
Simulation effects		4.05 ± 0.71
The simulations helped me build my nutrition assessment skills.	27 (79.1)	4.06 ± 0.69
The simulations helped me build my nutrition counseling skills.	30 (88.2)	4.18 ± 0.63
The simulations helped me build my MNT knowledge level.	22 (64.7)	3.71 ± 0.97
The simulations helped me better understand professional behaviors and actions.	32 (94.1)	4.24 ± 0.55
Simulation guidance		
There was an opportunity to get guidance and feedback from the instructor to build my knowledge and skills.	29 (85.3)	4.24 ± 0.92
VR Simulation Experience		
Simulation realism		4.41 ± 0.55
The patient scenario felt realistic.	33 (97.1)	4.32 ± 0.54
The emotions of the characters in the simulation felt like what I might encounter as a dietitian.	33 (97.1)	4.50 ± 0.56
Agreeableness of the simulation		
The simulation was a helpful learning experience.	34 (100.0)	4.47 ± 0.51
I enjoyed the simulation.	33 (97.1)	4.62 ± 0.65
Anticipated effects of the simulation		4.50 ± 0.60
The simulations will help me do well in a clinical setting.	32 (94.1)	4.47 ± 0.62
The simulation will help me when I work with real patients.	33 (97.1)	4.53 ± 0.56
The simulation will help me better manage patient emotions.	32 (94.1)	4.50 ± 0.62
VR Experience would benefit…		
Dietetic students earlier in undergraduate program	24 (70.6)	-
Dietetic students in their internship before doing clinical rotations [supervised practice]	30 (88.2)	-
Dietetic students in their internship after doing clinical rotations [supervised practice]	14 (41.2)	-
Other medical professionals	17 (50.0)	-

* MNT = Medical Nutrition Therapy.

## Data Availability

Data can be made available upon request of the corresponding author. The questionnaires used for this study can also be made available upon request of the corresponding author. The data are not publicly available due to privacy reasons.

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
