# Peer review of "iENDEAVORS: Development and Testing of Virtual Reality Simulations for Nutrition and Dietetics"

_ijerph, 2025, doi:10.3390/ijerph22091389_

Round 1
Reviewer 1 Report
Comments and Suggestions for Authors
I enjoyed the opportunity to review this paper. This application of VR is novel and much needed to catch dietetics up with other health professionals education. Below are a few suggestions that are mostly points of clarification.
Line 73 – Since these educational uses of VR haven’t been tested (until your study), I would suggest moderating the langue here to reflect more potential applications of VR and describe more specifically how the immersive and experiential nature of VR education could possibly accomplish these outcomes. Also make the comparison to what kind of traditional education more clear – lectures, case studies, etc.
Line - Reference 29 refers to “patient” nutrition education, not dietetics education for future professionals. VR programs for nutrition education or nutrition related behaviors are ahead of the use for professional education in the dietetics field. Maybe adjust this statement to say as much? Is there another citation that could be used? If this is the first application, maybe there are no references? Could that be stated here – that this is the first (if that is true)?
Materials and Methods:
Line 124 – more detail on the process for selection of these skills, are they most critical to competency, most difficult to achieve through traditional methods, etc?
Table 1. remove tab after bullet points in the learning objectives column
Line 157 – What specific meta platform is used? Is this meta quest that is commercially available? Is line 159 stating that the software is able to be used on multiple VR devices?
Line 163 – What is meant by specific interface for interaction with the software?
For this paragraph- photos of the characters would be helpful to understand what is being described.
Line 181 – formative testing, any data on “user acceptance”
Line 197 – image here would help
Line 199 – describe this more, is this chat bot that responds to the user or is it point and click options
The detail description of the program development is appreciated and provides transparent picture of how decisions were made. A suggestion to help the reader follow would be to perhaps condense or streamline some of this discussion. One thought would be to separate out descriptions of the process from descriptions of the program itself. Maybe add a section that walks the reader through the user experience (bits and parts of this are already sprinkled throughout)
Line 238 – Pilot testing: my understanding is the pilot testing was used to make some inital improvements to the program. To make this more clear, maybe state this at the begining of this section.
Enjoyed reading the qualitative results of the pilot study, but if possible this section could be condensed to a table of main themes or needed improvements to the program and more detail reported out in an appendix.
Line 365 – are users speaking?
Line 391 – Is this knowledge related to what? Clear once see Table 4- but maybe just clarify a bit here
Could the questionnaires be included as appendices
Line 425 – Clarify that this questionnaire is specifically related to the simulations. At first read I wasn’t sure if this was the same pre-test questions being reported as before, during and after. Once I read Table 5 it was clear.
Table 3 – font in the footnotes
Table 4 – could just report the P value for t-test?
Line 494 – instructor within the VR ?
Throughout the discussion, can more comparison to literature (other fields that use VR simulations learning or learning theory)
Line 532 – any learning theory that speak to this
Line 538 – is the qualitative feedback from the pilot test only? If yes, specify and if no, can the qualitative feedback be reported in results ?
Line 555- 556 can this statement be cited?
Line 560 – debriefing was this within the VR or part of the study procedures
Author Response
Article Title: iENDEAVORS: Development and Testing of Virtual Reality Simulations for Nutrition and Dietetics
Response to Reviewer #1 Comments
Author’s Response: Thank you very much for taking the time to review this manuscript. Please find the detailed responses below and the corresponding revisions/corrections highlighted using track changes in the re-submitted files to be considered for publication in this journal.
Comment: I enjoyed the opportunity to review this paper. This application of VR is novel and much needed to catch dietetics up with other health professional education. Below are a few suggestions that are mostly points of clarification.
Response: Thanks. Your suggested revisions were helpful with further improving our manuscript.
Comment: Line 73 – Since these educational uses of VR haven’t been tested (until your study), I would suggest moderating the language here to reflect more potential applications of VR and describe more specifically how the immersive and experiential nature of VR education could possibly accomplish these outcomes. Also make the comparison to what kind of traditional education more clear – lectures, case studies, etc.
Response: Agree! We have revised the language as requested as shown in lines 71 to 79.
Comment: Line - Reference 29 refers to “patient” nutrition education, not dietetics education for future professionals. VR programs for nutrition education or nutrition related behaviors are ahead of the use for professional education in the dietetics field. Maybe adjust this statement to say as much? Is there another citation that could be used? If this is the first application, maybe there are no references? Could that be stated here – that this is the first (if that is true)?
Response: Thank you for pointing this out. To our knowledge, there are no other VR simulations for dietetics education that have been developed and tested for their effectiveness in dietetics education in students. We have revised lines 99 - 101 and removed the in-text citation (29), as suggested.
Comment: Line 124 – more detail on the process for selection of these skills, are they most critical to competency, most difficult to achieve through traditional methods, etc?
Response: As requested, we have provided some more detail to the process of selection of these skills which can be found in the revised lines 131 – 137.
Table 1. remove tab after bullet points in the learning objectives column
Response: Thanks for pointing this out. I believe the formatting of the table was slightly altered when imported to the journal’s template. Table 1 is now revised as suggested.
Comment: Line 157 – What specific meta platform is used? Is this meta quest that is commercially available? Is line 159 stating that the software is able to be used on multiple VR devices?
Response: This question helped us realize we were not complete in listing all of the software used for development, and may cause confusion on the part of the reader at worst, and provide unnecessary information at least. We decided to focus instead on the delivery platform (the system users need to access the program) rather than provide details on all of the software and versions used for modeling, texturing, animating, coding, script-writing, audio editing and managing (which may be more appropriate for a technical article on the development of the software). We have clarified platform and device for use (Meta Quest) in lines 165-173.
Comment: Line 163 – What is meant by specific interface for interaction with the software?
For this paragraph- photos of the characters would be helpful to understand what is being described.
Response: To clarify, the VR interface refers to the set of elements users interact with to navigate and control their experience within the VR environment. Lines 175-176 were revised for clarification.
Comment: Line 181 – formative testing, any data on “user acceptance”
Response: The creation of the characters was formatively tested internally by the research team so no data was collected. This sentence was slightly rephrased to “…determine potential user acceptance,… (Line 194).
Comment: Line 197 – image here would help
Response: Our study program website provides images of the four characters for viewing as well as a short video clip for an example of the user experience. A sentence was added to this paragraph to refer readers to our study program website (lines 211-213).
Comment: Line 199 – describe this more, is this chat bot that responds to the user or is it point and click options
Response: This refers specifically to an audio file that was recorded with animation created to match as a test for the emotional capacity of developed animation. Thank you for clarifying the question about chat bots, however. We took the opportunity to clarify the genesis of the interactions script in the paragraph starting in line 221 and emphasized that the reactions of the characters were scripted, and not AI generated.
Comment: The detail description of the program development is appreciated and provides transparent picture of how decisions were made. A suggestion to help the reader follow would be to perhaps condense or streamline some of this discussion. One thought would be to separate out descriptions of the process from descriptions of the program itself. Maybe add a section that walks the reader through the user experience (bits and parts of this are already sprinkled throughout)
Response: Thank you for this suggestion. Given the amount of time and effort that was dedicated to developing the VR simulations, we feel strongly that it is important to be detailed and transparent about the decisions that were made so others may be more aware of what all goes into this process. If the editor feels strongly that we need to streamline the amount of information provided due to page limits, we can certainly make these suggested revisions.
Comment: Line 238 – Pilot testing: my understanding is the pilot testing was used to make some initial improvements to the program. To make this more clear, maybe state this at the beginning of this section.
Response: Thanks for pointing this out. To clarify, we have added a statement to reflect that pilot testing was performed to further improve the VR simulations prior to field testing on lines 257-258.
Comment: Enjoyed reading the qualitative results of the pilot study, but if possible this section could be condensed to a table of main themes or needed improvements to the program and more detail reported out in an appendix.
Response: Thank you for this suggestion. We have tried to create a table as suggested, but found that it was unable to sufficiently capture the important details, especially those that could help facilitate the work of future developers.
Comment: Line 365 – are users speaking?
Response: To clarify, the users do not speak during the VR simulations. They are sitting and listening to the conversations between the RDN and patient and interacting with their hands in the VR space. Line 400 was slightly revised to make this more evident.
Comment: Line 391 – Is this knowledge related to what? Clear once see Table 4- but maybe just clarify a bit here
Response: Thanks for pointing this out. We have revised this sentence to show that the two multiple choice items per VR simulation were knowledge-related questions specific to each scenario (Lines 418-419).
Comment: Could the questionnaires be included as appendices?
Response: The questionnaires are very specific to the simulations. We have added a note at the end under data availability section on how others may be able obtain them from the corresponding author.
Comment: Line 425 – Clarify that this questionnaire is specifically related to the simulations. At first read I wasn’t sure if this was the same pre-test questions being reported as before, during and after. Once I read Table 5 it was clear.
Response: Good point! For clarification, this sentence was revised to state “During the post-test survey only, …” (Line 453).
Comment: Table 3 – font in the footnotes
Response: The formatting in Table 3 footnote was revised, as suggested.
Comment: Table 4 – could just report the P value for t-test?
Response: In being transparent with the results from our data analyses, we feel it is important to include all information provided in Table 4, including the effect sizes (Cohen’s d values).
Comment: Line 494 – instructor within the VR ?
Response: To clarify, in this context, the instructor is referring to the research assistants who were standing by to assist users if they ran into problems navigating the VR simulation during the field testing. This sentence was revised to be more clear for readers to interpret (see lines 531-532).
Comment: Throughout the discussion, can more comparison to literature (other fields that use VR simulations learning or learning theory)
Response: Thanks for this suggestion. As much as possible, we tried to incorporate comparisons of our results to other fields that use VR simulations.
Comment: Line 532 – any learning theory that speak to this
Response: Thanks for pointing this suggestion. We have revised lines 571-574 to reference to the importance of active learning methods in teaching that is provided by VR simulations.
Comment: Line 538 – is the qualitative feedback from the pilot test only? If yes, specify and if no, can the qualitative feedback be reported in results ?
Response: To clarify, the qualitative feedback mentioned in the discussion is referring to data collected in the pilot test study. This sentence was revised on line 579 to make this clear to the reader.
Comment: Line 555- 556 can this statement be cited?
Response: As requested, this sentence was cited (line 598).
Comment: Line 560 – debriefing was this within the VR or part of the study procedures.
Response: The debriefing was held outside the VR simulation with a trained facilitator as discussed in the field test design section.
Reviewer 2 Report
Comments and Suggestions for Authors
This paper covers an interesting topic. VR has potential to markedly change nutrition/dietetics education but there appears to be a limited research base to prove its effectiveness. This paper describes the design of several VR simulations, revisions based on feedback and field testing to determine feasibility and acceptability. There are a lot of strengths to this study - the team based approach is a particular strength. However. I would like to see much more information about how decisions were made, some consideration about the statistical analysis (control for multiple comparisons), and to be more circumspect about the interpretation of the results considering the lack of a control group. My specific comments are:
More information on the processes used to identify the skills that are challenging to demonstrate and evaluate would be useful - what was considered? By what criteria were they evaluated and why did these skills be deemed the highest priority?
The same for the selection of VR equipment - what criteria was used to determine that Oculus Quest was the best platform?
Which version of Unity was used for the development?
Who determined that the characters were in the uncanny valley? Was this based on feedback, pilot testing or the opinion of the researchers?
Were the 3D models created from a rendering package or pre-purchased? What package was used or what asset store was used to create the 3D models?
How was it confirmed that there was a strong emotional impact of the characters? Was any formal methods conducted to confirm this?
Was any assets used to help animate the characters? If so, these should be listed.
Picture od f the VR scenes and characters would be very useful to visualize what the participants saw.
For the pilot testing, were people randomized to simulations order? Did they choose preset prompts (and was this randomized?) or did they choose for themselves?
For pilot testing and field testing, were participants told the purpose of the study? How were the risks for a demand bias controlled?
Was a power calculation performed to justify the sample size used?
Many t-tests were performed - was a control for the inflation of type I error used?
The most serious issue with this paper is that there is no control group or comparison group. It is not possible to attribute the changes in outcomes to the VR simulation. Without a control group, causal inferences are unwarranted (for instance (line 516-518).
line 526-536 - a strong possibility is that VR, while cool, is not an effective medium? In the introduction the authors cite studies that support the use of VR in education but there are also studies which have found no improvement. Could you put these previous studies into context with regards to your data? What were the effect sizes of these studies?
I am sympathetic to the argument that VR could provide a low stakes method to allow students to repeatedly practice a scenario. But why does VR have an advantage over other methods? Traditional computers for instance. These would not require investment in VR equipment but give the same result.
I may have missed this but a significant barrier to VR use is cybersickness - was this assessed? I suggest it merits discussion in the paper.
Author Response
Article Title: iENDEAVORS: Development and Testing of Virtual Reality Simulations for Nutrition and Dietetics
Response to Reviewer #2 Comments
Author’s Response: Thank you very much for taking the time to review this manuscript. Please find the detailed responses below and the corresponding revisions/corrections highlighted using track changes in the re-submitted files to be considered for publication in this journal.
Comment: This paper covers an interesting topic. VR has potential to markedly change nutrition/dietetics education but there appears to be a limited research base to prove its effectiveness. This paper describes the design of several VR simulations, revisions based on feedback and field testing to determine feasibility and acceptability. There are a lot of strengths to this study - the team based approach is a particular strength. However. I would like to see much more information about how decisions were made, some consideration about the statistical analysis (control for multiple comparisons), and to be more circumspect about the interpretation of the results considering the lack of a control group. My specific comments are:
Response: Thanks. Your constructive feedback has been helpful with further improving our manuscript.
Comment: More information on the processes used to identify the skills that are challenging to demonstrate and evaluate would be useful - what was considered? By what criteria were they evaluated and why did these skills be deemed the highest priority?
Response: As requested, we have provided some more detail to the process of selection of these skills which can be found in the revised lines 131 – 137.
Comment: The same for the selection of VR equipment - what criteria was used to determine that Oculus Quest was the best platform?
Response: As requested, we have provided more language on the process of selection of the VR equipment on lines 165 - 173.
Comment: Which version of Unity was used for the development?
Response: Thank you for asking. This question helped us realize we were not complete in listing all of the software used for development, and may cause confusion on the part of the reader at worst, and provide unnecessary information at least. We decided to focus instead on the delivery platform (the system users need to access the program) rather than provide details on all of the software and versions used for modeling, texturing, animating, coding, script-writing, audio editing and managing (which may be more appropriate for a technical article on the development of the software). We have clarified platform and device for use (Meta Quest) in lines 165-173.
Comment: Who determined that the characters were in the uncanny valley? Was this based on feedback, pilot testing or the opinion of the researchers?
Response: To clarify, the research team perceived the characters as falling into the uncanny valley. Lines 180-182 was revised to make this statement clearer.
Comment: Were the 3D models created from a rendering package or pre-purchased? What package was used or what asset store was used to create the 3D models?
Response: All of the models are original and custom to the design team. We have added the word “original” in line 192 to indicate this. Also, in lines 199-201 we add a sentence to further clarify that the developers created the models themselves.
Comment: How was it confirmed that there was a strong emotional impact of the characters? Was any formal methods conducted to confirm this?
Response: To clarify, the multidisciplinary research team members provided feedback as the character animations were being developed in the effort towards wanting to create characters with a strong emotional impact. There were no formal methods conducted to confirm this during the development phase. Lines 194 to 195 were revised to be moderate the tone of this statement.
Comment: Was any assets used to help animate the characters? If so, these should be listed.
Response: Thank you for clarifying this question. I believe in “assets” you must indicate pre-rendered models or purchased assets were used. The term “original” has been added in line 192 to clarify that the work is original to the team.
Comment: Picture of the VR scenes and characters would be very useful to visualize what the participants saw.
Response: Our study program website provides images of the four animated characters for viewing as well as a short video clip for an example of the user experience. A sentence was added to lines 211-213 to refer readers to our study program website.
Comment: For the pilot testing, were people randomized to simulations order? Did they choose preset prompts (and was this randomized?) or did they choose for themselves?
Response: Thanks for pointing this out. For the pilot test, participants were randomly assigned a VR simulation. During the VR simulation, participants could choose their own responses to the prompts provided. Lines 267 – 270 were revised to make this more clear.
Comment: For pilot testing and field testing, were participants told the purpose of the study? How were the risks for a demand bias controlled?
Response: For pilot testing and field testing, all participants were told the purpose of the study (see lines 271 – 273 and lines 392 – 396). To avoid study biases as much as possible, a standardized protocol was used with trained research assistants.
Comment: Was a power calculation performed to justify the sample size used?
Response: An a priori power analysis was performed to justify our sample size in the study. This statement has been added to the “Data Analysis and Power Sample” section in lines 476 – 480.
Comment: Many t-tests were performed - was a control for the inflation of type I error used?
Response: Thanks for pointing this out. We have revised our data analysis (lines 472 – 474) and result sections of the manuscript by using the Benjamini-Hochberg procedure (also known as the False Discovery Rate) to correct for multiple comparisons in the paired-tests performed as shown in the revised Table 4.
Comment: The most serious issue with this paper is that there is no control group or comparison group. It is not possible to attribute the changes in outcomes to the VR simulation. Without a control group, causal inferences are unwarranted (for instance (line 516-518).
Response: We agree that having a control group would have been advantageous. However, due to the challenges with recruitment and lacking a suitable control condition, we were unable to proceed with a control group. We have moderated our language in the discussion and made sure to highlight a major study limitation with lack of a control group in lines 620 - 621.
Comment: Line 526-536 - a strong possibility is that VR, while cool, is not an effective medium? In the introduction the authors cite studies that support the use of VR in education but there are also studies which have found no improvement. Could you put these previous studies into context with regards to your data? What were the effect sizes of these studies?
Response: Thank you for this comment. We are not aware of VR simulations in dietetics to which we can compare our results. If there are pertinent examples in other fields the reviewer could suggest, we are happy to compare results.
Comment: I am sympathetic to the argument that VR could provide a low stakes method to allow students to repeatedly practice a scenario. But why does VR have an advantage over other methods? Traditional computers for instance. These would not require investment in VR equipment but give the same result.
Response: VR simulations are not meant to replace traditional methods but to enhance experiential learning for students, especially when preparing them for entry into novel roles and situations such as functioning as a professional in health care settings. VR simulations may be especially helpful for dietetic interns that need repeated exposure to further build their nutrition counseling skills prior to interacting with real patients in a facility.
Comment: I may have missed this but a significant barrier to VR use is cybersickness - was this assessed? I suggest it merits discussion in the paper.
Response: In our study protocol for the pilot testing and field testing of VR simulations, participants were informed that if they felt dizzy or nauseous at any time, they could stop the VR simulation. Participants were also seated during the simulation with a time limit of 30 minutes to help prevent cybersickness. These precautions appeared to have worked because none of the participants in our study reported feeling dizzy or nauseous while participating. A few sentences were added in lines 404-406 to address this concern.
Round 2
Reviewer 2 Report
Comments and Suggestions for Authors
The authors have responded carefully to my comments and I appreciate their work. I recommend the paper is accepted and wish them well with their work in this area!